# Liver Transplant Oncology: Towards Dynamic Tumor-Biology-Oriented Patient Selection

**DOI:** 10.3390/cancers14112662

**Published:** 2022-05-27

**Authors:** Matthias Ilmer, Markus Otto Guba

**Affiliations:** 1Department of General, Visceral and Transplantation Surgery, Hospital of the University of Munich, Ludwig-Maximilians-University (LMU), 81377 Munich, Germany; markus.guba@med.uni-muenchen.de; 2German Cancer Consortium (DKTK), Partner Site Munich, German Cancer Research Center (DKFZ), 69120 Heidelberg, Germany; 3Transplantation Center Munich, Ludwig-Maximilians-University Munich, Campus Grosshadern, 81377 Munich, Germany; 4Liver Center Munich, Ludwig-Maximilians-University Munich, 81377 Munich, Germany

**Keywords:** transplant oncology, HCC, liver transplant, tumor biology, immunotherapy

## Abstract

While liver transplantation was initially considered as a curative treatment modality only for hepatocellular carcinoma, the indication has been increasingly extended to other tumor entities over recent years, most recently to the treatment of non-resectable colorectal liver metastases. Although oncologic outcomes after liver transplantation (LT) are consistently good, organ shortage forces stringent selection of suitable candidates. Dynamic criteria based on tumor biology fulfill the prerequisite of an individual oncological prediction better than traditional morphometric criteria based on tumor burden. The availability of specific (neo-)adjuvant therapies and customized modern immunosuppression may further contribute to favorable post-transplantation outcomes on the one hand and simultaneously open the path to LT as a curative option for advanced stages of tumor patients. Herein, we provide an overview of the oncological LT indications, the selection process, and expected oncological outcome after LT.

## 1. Introduction

Liver transplantation (LT) has become a standard procedure for the treatment of acute and end-stage liver failure as well as for numerous oncological indications. Not only are the results for benign transplant indications excellent, but also oncologic outcomes far exceed those of alternative non-transplant treatment modalities.

The major but critical hurdle to a wider use of liver transplantation in oncology is organ shortage. Consequently, the indication must be limited to those patients who benefit most from liver transplantation, if only to give other non-oncological patients a chance for liver transplantation. In this regard, selection criteria for assessing the transplant benefit in oncological liver transplantation are of particular importance. The requirement for these criteria is that they reflect the individual oncological outcome after LT as accurately as possible, are easy to determine in routine clinical practice, and can be audited for quality assurance.

Within the traditional concept, the expected outcome is extrapolated over the tumor burden. Tumor burden, in turn, is determined as a function of the number of nodes and tumor size, largely disregarding individual tumor biology (Figure 1).

In a large number of studies, this tumor burden-based concept has been repeatedly challenged due to poor discrimination between good and bad performers. Recognizing this distinctive weakness of morphometric criteria, elements of individual tumor biology assessment have been increasingly incorporated into selection algorithms. In this context, not the static but rather the longitudinal assessment of the individual tumor biology, including by stable response to neoadjuvant therapy, seems to be of particular importance, imposing a dynamic selection process. Within the framework of this concept, which is also supported by the real course of disease in tumor patients, it is possible that patients with a high tumor burden but favorable tumor biology perform better than patients with a low tumor burden but unfavorable tumor biology. Overall, individual tumor biology appears to exert greater leverage than tumor burden in this regard. To illustrate this idea more clearly, we summarize the hypothesis in Figure 2.

The crucial questions here are how to determine individual tumor biology and how accountable the selection process is in a highly regulated allocation process.

In this review, we summarize present-day oncologic indications for liver transplantation. We review the currently applied and evolving selection criteria for oncologic liver transplant patients and implications for (neo-)adjuvant therapy and discuss the allocation equity issues that an expansion of oncologic indications in liver transplantation may entail.

## 2. Transplantation for Cancer

### 2.1. Hepatocellular Carcinoma (HCC)

HCC is the most common malignant primary liver tumor in adults, and the majority of cases arise in cirrhotic livers. Overall, the incidence is increasing worldwide. While in the past mainly viral hepatitis and alcoholic steatohepatitis were responsible, non-alcoholic steatohepatitis (NASH) has recently become a major contributor to an increasing incidence—especially in industrialized countries [1].

Due to compromised liver reserve, options for major liver resection are limited and liver transplantation remains the only option to radically remove the tumor and simultaneously restore liver function. Furthermore, from an oncologic perspective, liver resection and alternative local ablative procedures are burdened with high recurrence rates, primarily due to de novo tumors arising in the precancerous liver. Overall, about 70% of resected patients will present with tumor recurrence within 5 years [2]. In contrast, tumor recurrence rates after transplantation are comparatively low at 15–20% after 5 years, depending on patient selection [3].

Historically, selection of transplant candidates has been performed using static morphometric criteria. Mazzaferro et al. established the *Milan criteria* (MC) for selection of appropriate transplant candidates [4]. In this landmark study, patients with liver cirrhosis predictively survived better after LT if the postulated *MC* (one tumor up to 5 cm diameter or up to three tumors with none exceeding 3 cm) were fulfilled. Tumor biology and its potentially prognostic value plays no role in these still widely used criteria. 

However, the *Milan criteria* were found to be overly restrictive; hence, a multitude of extended morphometric criteria such as the UCSF criteria [5] with one solitary tumor <6.5 cm or up to three nodules <4.5 cm and a total tumor diameter of <8 cm were formulated with comparable oncological outcomes [5,6]. In attempts to identify patients that might benefit from LT outside *MC* and to respect improvements made in the diagnostics of smaller lesions, in 2009, Mazzaferro and colleagues extended their conditions from the very dichotomous nature of Milan to the so-called “*up-to-seven criteria*”, where HCCs were included with seven as the sum of the size of the largest tumor in cm plus the number of tumors. Patients after LT with HCCs fulfilling these criteria achieved excellent 5-year overall survival of 71.2% [6]. 

Although it was recognized early that the importance of tumor biology, such as a poor degree of differentiation [7] or the state of microvascular invasion (MiV) [8], could serve as prognosis factors, tumor-biology-oriented factors have been incorporated only recently into selection algorithms. Interestingly, another landmark study by Mazzaferro et al. in Lancet Oncology 2009 showed that MiV within the *MC* influenced 5-year overall survival only marginally, but within the extended “*up-to-seven criteria*”, presence of MiV was associated with a reduced 5-year overall survival of 47.4% compared to 71.2% without MiV after LT [6]. More recently, prediction of MiV has been increasingly evaluated and detected by radiological features, e.g., certain defined MRI criteria, such as tumor size (>5 cm), rim arterial or peritumoral enhancement, peritumoral hypointensity, nonsmooth tumor margin, multifocality, and hypointensity on T1-weighted imaging [9,10]. However, some of the mentioned prognostic biological factors can only be unerringly determined with high precision after completion of histopathology of the explanted liver or with considerable expenditure in preoperative biopsies and are therefore not yet suitable for patient selection before LT. 

On the other hand, biomarkers such as AFP or dynamic criteria such as response to therapy are feasible to collect in routine clinical practice. In this regard, pretransplant AFP, representative for biochemical tumor burden, was shown to correlate well with posttransplant recurrence. Here, AFP was demonstrated to be an individual predictive factor for recurrence after LT [11]. Intriguingly, in the case of low pretransplant AFP (<10 ng/mL), it was shown that even HCC patients with radiological signs of vascular invasion might be candidates for LT, if those lesions were successfully treated [12]. In addition to AFP, a number of other biomarkers have been proposed but are not widely adopted, including des-γ-carboxyprothrombin (prothrombin induced by vitamin K absence-II (PIVKA-II)) [13], neurophil-to-lymphocyte ratio [14], or more generally, cancer-related symptoms (e.g., weight loss, fatigue).

Response to therapy as a selection criterion was first described by Otto et al. [15]. In the following, most studies used the *MC* as the target criterium [16]. Locoregional therapies, such as radiofrequency ablation (RFA) or transarterial chemoembolization (TACE) can be applied for successful downstaging. In a prospective randomized Italian multicenter trial, LT after downstaging showed significantly improved OS and tumor-free survival compared to the non-LT control [17]. Success rates for downstaging are acceptable and can be predicted by normalization of AFP levels prior to LT as well as wait times longer than 12 months [18]; recurrence rates after LT are satisfactory. However, tumor progression during waiting time for LT despite locoregional bridging appears to be an independent risk factor for increased recurrence and decreased outcome as shown in the SiLVER study [19]. Importantly, sufficiently long observation periods to identify favorable tumor biology need to be provided for successful LT outcomes [20,21,22]. Nonetheless, use of the *MC* in downstaging seems arbitrary, whereas relative grade of response, and more so stability towards downstaging over time, could serve as much more reliable and biologically dynamic criteria. This has also been shown in a recent large analysis of the US Multicenter HCC Transplant Consortium [23].

Our group has established a dynamic selection process based on three variables (AFP, response to locoregional therapy, waiting time >6 months) reflecting tumor biology, when extrahepatic disease and macrovascular invasion was excluded. In this cohort, 1- and 5-year overall survival rates were 91.1 and 73.9%, respectively, for patients fulfilling the *MC*; for patients outside the *MC* but selected by the abovementioned dynamic selection process, we found 86.7% and 71.7% OS rates at 1 and 5 years [24].

Recurrence of HCC after LT occurs in about 16% of patients; here, biological factors also determine the prognosis, e.g., early recurrence (potentially due to circulating tumor cells (CTCs) or undetected metastasis) vs. late recurrence (either by better containment of CTCs and occult metastasis or by de novo tumors in the newly grafted liver) as well as localized and isolated vs. multifocal (metastatic) disease. Surgical resection in these cases appears to be advantageous; theoretically, re-transplantation might be an option for biologically selected patients. If patients cannot be resected, loco-regional therapy is the best option, and systemically a combination of sorafenib with mTOR inhibitors improves patient survival [25]. Sorafenib-tolerant patients could further be treated by regorafenib, a multikinase inhibitor similar to sorafenib with a wider range of kinase inhibition modulating the tumor microenvironment and potentially promoting anti-tumor immunity. This might then extend overall survival in patients after HCC recurrence after LT without the option to resect [26].

Lastly, preoperative assessment of liver tumors and outcome after LT is still challenging, and sometimes even the distinction between benign and malignant lesions remains a demanding task. In this regard, the use of the Metroticket 2.0 calculator might help to assess preoperative HCC tumor burden and biology; however, a better assessment applying the Liver Imaging Reporting and Data System (LI-RADS) protocol has been recently suggested [27]. In particular, LI-RADS4 and 5 highly correlated with HCC pathology in examinations of explanted livers [28]. In this setting, it would be desirable to establish extended criteria of LI-RADS that could help to not only categorize the likelihood of finding HCC but also to evaluate biological features of HCCs, which could further help to estimate the use of LT as well as long-term prognosis. In this context, application and incorporation of artificial intelligence with a focus on radiomics in the process of finding the right candidate and maybe the right organ could be of prime value in transplant oncology.

### 2.2. Cholangiocellular Carcinoma (CCA)

Cholangiocellular Carcinoma (CCA) is an entity of very aggressive adenocarcinomas that derive from the biliary tree epithelium and can be subdivided by their anatomical site into intrahepatic (~5–10%) and extrahepatic with further subdivision of the latter into perihilar (~50–60%) and distal (~20–30%) CCAs. Although rare, it is the second most common primary liver cancer after HCC. The gold standard of treatment is surgical resection with 5-year overall survival for perihilar CCA (phCCA) of 25–40% [29]; however, technical, e.g., locally advanced or metastasized stages, as well as functional irresectability (e.g., parenchymal hepatic changes as in primary sclerosing cholangitis (PSC)), limit curative surgical approaches to a minor percentage of CCA patients [30]. In 2005, Rea and colleagues were able to show that LT for perihilar CCA could be a veritable option with acceptable long-term survival and 5-year disease free survival rates of ~65%. This study could show that LT was superior to resection with the limitation that the transplantation group was younger and with higher incidences of PSC and/or inflammatory bowel disease. Neoadjuvant chemotherapy was applied to all LT patients [31]. Since then, the Mayo protocol has been further refined as outlined below. 

In a large international multicenter cohort, LT for very early intrahepatic CCA (iCCA) in cirrhotic livers as defined by single tumors comprising a diameter of <2 cm was retrospectively shown to be significantly better than in advanced tumors (>2 cm or multifocal) [32]. The 5-year OS was 65%, and risk of recurrence was 18%. However, the main limitation of this study was that CCA was detected as incidentaloma during cirrhosis workup or was mistaken for HCC. Risk factors for recurrence identified in multivariate analysis of this study were poor tumor differentiation and presence of MiV. Prospective evaluation of such an LT approach is currently being investigated in ongoing Canadian and Norwegian studies (Table 1).

Patients with locally advanced intrahepatic CCA without cirrhosis plus response or stability towards neoadjuvant gemcitabine-based chemotherapy were shown to benefit from LT in 2018 [33]. Lunsford and colleagues found in this prospective study that by fulfilling the dynamic biologic criteria as potential surrogates of favorable tumor biology, namely at least 6 months’ radiological response or stability after neoadjuvant chemotherapy, 5-year OS reached 83.3% and DFS 50%. Remarkably, three out of six patients received diseased organs from donors with extended criteria which would not have been transplanted otherwise. A recent update from the McMillan and colleagues showed that even advanced iCCAs with a median tumor size of 10.4 cm and a median number of nodules of two might also benefit from LT. Here, favorable genetic alterations (e.g., in FGFR and DNA damage repair pathways) that associated iCCAs with lower aggressive behavior and superior response to neoadjuvant therapy might add increased value to the selection process. However, recurrence occurred nevertheless in almost 50% of transplanted patients [34].

To generate more meaningful data, it is generally recommended to transplant phCCAs within defined clinical trial protocols. According to the Mayo criteria, morphometric measures include tumor diameter <3 cm and origin above the cystic duct. Dynamic criteria include no evidence of extrahepatic or lymphatic spread, pretreatment with neoadjuvant therapy (e.g., chemo-irradiation), and confirmation by tumor biopsy or radiologically malignant-appearing stricture with a CA19-9 level >100 U/mL [35]. The tumor must be technically unresectable due to extensive vascular and/or biliary invasion or functionally unresectable due to poor hepatic functional reserve for an underlying liver disease which might predispose the patient to post-hepatectomy liver failure [29]. However, these criteria remain a matter of debate. Recently, the (un)resectability of phCCAs defined by the abovementioned criteria has been challenged, while outcome (5-year OS) of resected patients in a different cohort was 67.1%, questioning the advantage of LT over resection [36].

In order to collect objective criteria and for a better understanding of LT in phCCA as well as refinement of indications, multiple prospective studies are currently ongoing. We summarize these studies on LT in phCCAs in Table 1. Several of these prospective studies are in the process of recruitment, most notably to evaluate the use of LT in phCCAs in comparison to conventional resection for resectable (TRANSPHIL) or unresectable phCCAs (TESLA-II).

Selection of ideal candidates for LT in the case of iCCA is certainly more challenging and remains controversial. Early stage iCCA with a diameter smaller than 2 cm in cirrhotic livers or advanced iCCA in non-cirrhotic livers might benefit from upfront LT. In the dynamic biological sense of applied criteria, stability towards chemotherapy of more than 6 months should be expected [33,37]. Liver resection for iCCA remains the gold standard, and LT should be performed only after inclusion in studies with strict protocols [29]. Notable studies are underway in Norway and Canada (Table 1).

So far, reported outcomes in highly selected cases of mostly retrospective studies are promising. Early iCCAs with single tumors <2 cm in cirrhotic livers showed OS of 65–83% [32,38]. In non-cirrhotic livers, LT might be an option for advanced tumors without extrahepatic disease; however, very little is known about the outcomes at the moment. Currently ongoing studies (e.g., TESLA) might reveal a better profile of suitable candidates as well as outcomes. For phCCAs, strict inclusion criteria as well as adjusted selection of patients fulfilling strict criteria might lead to recurrence-free survival after LT of up to 72% [39].

### 2.3. Hepatic Epitheloid Hemangioendothelioma (HEHE)

Hepatic epitheloid hemangioendothelioma (HEHE) is a very rare vascular tumor of mesenchymal origin deriving from endothelial cells in the liver. The etiology of this sarcomatoid malignancy is unknown. In the liver, its growth pattern is usually multifocal (87%), invasive, and displacing; moreover, HEHE can form extrahepatic disease through lymphatic and hematogenous spread in ~37% of cases [40,41]. Long-term survival in HEHE patients is good with 5-year survival rates over 50% if adequately treated. Survival after transplantation is excellent and much better than HCC, CCA, and neuroendocrine liver metastases (NET-LM) [42]. First line surgical treatment is resection which is only possible in less than 10% of patients due to the multicentricity of HEHE lesions or technical unresectability [43].

As described for the other entities, a dynamic selection process to select patients for LT was recently proposed and outlined [40]. With the identified risk factors (macrovascular invasion, pre-LT waiting time <120 days, hilar lymph node invasion), stratification based on the so-called HEHE-LT score could be carried out in a low-risk and high-risk group and might be an interesting tool to select the ideal patient for LT. In this ELTR-ELITA registry analysis, pulmonary metastasis was no strict exclusion criteria when resectable in one case, even by lung transplantation [4,44].

Among the very heterogeneous group of primary hepatic vascular sarcomas, HEHE comprises an entity with rather indolent behavior with 5-year survival of 55–75% after either resection or transplantation. Whereas the more aggressive angiosarcoma seemed to be better treated by resection, in HEHE, both treatment options provided excellent outcomes [45]. Although HEHEs were larger and more likely to be node positive in the LT group, it is not clear at that moment whether the reported outcomes are due to a better treatment with LT or the good-natured behavior of the malignancy. A point of discussion remains in that transplantation might be an overtreatment for HEHE [46]. In that regard, large tumor size, extrahepatic disease beyond portal lymph nodes, as well as patient age were reported to negatively impact overall survival and could hence be taken into account for LT evaluation [47].

Overall, selection of the right candidate after stratification by, e.g., the abovementioned HEHE-LT score, should help to ensure best survival after transplantation and therefore justify fair distribution of limited organs.

### 2.4. Hepatoblastoma (HB)

Hepatoblastoma is the most common primary malignant liver tumor of childhood with an increasing incidence over recent years. At diagnosis, these tumors often present at an advanced stage, and some of them show macrovascular invasion or distant metastases. Risk stratification of tumor burden as well as prognosis is usually classified by the pretreatment extent of disease (PRETEXT) [48]. Assignment happens to four groups (I, II, III, IV) by the number of bordering involved sections. Furthermore, involvement of hepatic veins (V), portal vein (P), caudate lobe (C), extrahepatic adjacent (E), or distant metastasis (M) is labeled [49]. As most of the advanced tumors are treated by neoadjuvant chemotherapy, extension is assessed by post-treatment extent of disease (POST-TEXT) with similar annotations. Good responses can be achieved with platinum-based neoadjuvant regimens; children subsequently undergo curative liver resection, even in selected patients with POST-TEXT III or IV [50]. However, in some cases, despite chemotherapy, liver resection is technically not feasible, e.g., in central lesions, involvement of major vessels (P or V) or multifocal disease, leaving liver transplantation as the only remaining curative treatment option with long-term recurrence free survival (RFS) [51].

For the evaluation and selection of potential transplant candidates, extrahepatic disease is not necessarily a contraindication for LT. Up to 20% of HB patients present with metastasis to the lung at the point of diagnosis. First line treatment of metastatic disease includes aggressive chemotherapy; responsive behavior should lead to surgical clearance of the lung tumors in order to prepare patients for transplant when the liver tumors continue to fulfil transplant criteria [52]. In successfully transplanted patients, no relapse after three years was reported by the International Childhood Liver Tumor Strategy Group (SIOPEL) [53]. On the other hand, untreatable unresponsive or progressive metastatic disease should be considered a contraindication for LT, conditions that would also qualify as dynamic biologic criteria in our proposed model (Figure 2).

Recommendation for LT evaluation according to SIOPEL [53] should be considered, if HB is irresectable in central PRETEXT II/III with hepatic venous (V) or portal vein involvement (P), multifocality equivalent to PRETEXT IV or tumors unresponsive to chemotherapy (POST-TEXT IV or P2 or V3).

Outcome after LT in HB in general is excellent. Recurrence rates for HB are 0% for LT at 5 years and hence much better than after resection (~10%); 5-year survival was more than 80% [51,54]. Higher risk in HB is classified by the Children’s Hepatic tumors International Collaboration (CHIC) [55] by several parameters, including histology (pure fetal better than small cell undifferentiated), biological behavior (worse prognosis with lower AFP levels <100 U/mL, spontaneous tumor rupture at involvement or locally advanced tumors with macroscopic vascular invasion and metastatic disease), and age at time of diagnosis [56]. Older age, however, was reportedly associated with mixed hepatocellular carcinoma–hepatoblastoma histology and therefore presumably worse prognosis [57]. Unresponsiveness to chemotherapy was also accounted for with worse tumor biology leading to treatment with LT.

Although not at a significant level, 5-year OS was lower after LT in patients with metastatic disease in comparison to liver resection (67.3% vs. 80.7%) [56]. Research in this area is scarce, but a more detailed analysis would be desirable, especially with regards to the question of perioperative systemic therapy as well as (modified) immunosuppression to improve both transplant rejection issues as well as suppression of recurrent disease. In this regard, sirolimus-based immunosuppression seems safe and effective with improvements to renal function [58] by reducing calcineurin inhibitor doses in pediatric LT recipients [59]. This strategy might be promising as in switches to everolimus after chronic graft failure indicate [60,61]; however, general recommendations are yet to be made.

### 2.5. Neuroendocrine Liver Metastases (NET-LM)

Neuroendocrine tumors are a rare group of heterogeneous neoplasms that account for approximately 2% of malignancies of the gastroenteropancreatic (GEP) system. At the time of diagnosis, 40–50% of patients with pancreatic or small intestine NETs, the two most common GEP-NETs, already present with liver metastasis [62]. The only curative therapy is surgical resection of the primary tumor and liver metastasis. Most patients show disseminated liver disease not accessible by liver resection. In highly selected patients with no other metastatic disease other than hepatic involvement and after resection of the primary tumor, liver transplantation may be discussed as a treatment option.

Generally, only well differentiated NETs (G1/2) with Ki67 index <10% are eligible for liver transplantation, but recurrence rates after liver transplantation for NET-LM are high with up to 30–60% after 5 years [63]. To better select appropriate candidates, Mazzaferro and colleagues have established the so-called *Milan criteria* for NET-LMs allowing for excellent long-term outcome and low tumor recurrence rates [64].

However, the abovementioned retrospective cohort study is viewed critically, not only because the control group performed worse than expected under up-to-date therapy. The control group contained significantly older patients (15 years older) as well as more G2 tumors. These *MC* criteria for NET-LM [64] include patients who were 55 years of age or younger, whose primary tumor was a low-grade NET with a Ki67 index <10% draining through the portal vein system, with no more than 50% liver involvement, who responded to treatment, and whose disease had been stable for at least 6 months as stated in the ENETS guidelines [65].

In this context, it is not easy to identify those patients who will benefit from liver transplantation. On the one hand, the progression of NET-LM is much slower than in other GEP carcinomas and is well controlled by systemic therapies, such as somatostatin analogues (PROMID, CLARINET [66]), peptide receptor radionuclide therapy (PRRT) with 177-lutetium (NETTER-1 trial [67]) and the mammalian target of rapamycin inhibitor (mTOR) everolimus (RADIANT trials [68]) over a long period of time [67]; on the other hand, recurrence rates after liver transplantation are high, so transplantation as a curative therapy remains questionable. Therefore, the window for transplantation leading to a meaningful transplantation benefit is difficult to determine, especially because prospective and randomized clinical trials are missing [62].

### 2.6. Colorectal Cancer Liver Metastases (CRLM)

Colorectal cancer (CRC) is one of the most common tumor entities in Western countries, with approximately 70% of affected patients developing liver metastases, of which only about 20% are primarily resectable. With resection, 5-year survival rates of 30–40% can be expected, compared to only 5% in unresected patients. Already in the early days of liver transplantation, it was brought into play as a treatment option for patients with non-resectable liver metastases. However, due to the poor results with a 5-year survival of less than 20% and the scarcity of organs, this indication was quickly abandoned. It was not until the seminal work of the Oslo group (SECA-1 study) that this indication became acceptable again [69].

The first experience with liver transplantation showed a significant improvement of overall survival as compared to patients with palliative chemotherapy. The results, however, were burdened by a high tumor recurrence rate, raising the question whether liver transplantation for CRLM could be a curative treatment option. Meanwhile, the selection criteria are becoming increasingly refined, and in highly selected patients recurrence rates are low. Important selection criteria include mainly dynamic parameters, such as response to chemotherapy and no extrahepatic disease but also tumor biological parameters with regards to typical mutations (BRAF^WT^, KRAS^WT or MT^) [70]. This is also part of ongoing study protocols (Table 1).

The time from diagnosis to liver transplantation appears to have a high impact on the posttransplant survival rates, suggesting a natural selection of those tumors with favorable biology. In this regard, the Oslo group designed the prospective SECA-2 study. Candidates for LT needed to respond to chemotherapy with at least 10% according to RECIST criteria from the point of diagnosis to the time for LT, and time from diagnosis to LT mandatorily exceeded 1 year [71]. Toso et al. reported 1- and 6-year survival rates of 84% and 50%, respectively, with disease free survival rates of 56% and 38% at 1 and 5 years [72].

With the more refined criteria, outcomes are increasingly becoming better. Interestingly, clinical scores, such as the Fong or Oslo scores, as well as metabolic tumor volume seem to impact DFS and OS. Low scores were associated with at least 67% 5-year OS and significantly improved DFS compared to high scores [71,73]. According to the importance of dynamic selection biology (Figure 2), patients with high tumor burden but low biological relevance might benefit the most from LT compared to resection (5-year OS 69.1% vs. 14.6%, 1-year DFS 54.2% vs. 11.5%) [74].

In most health care systems, scarcity of organs limits higher recruitment numbers of patients for LT in CRLM. To circumvent this issue, a novel concept called resection and partial liver segment 2/3 transplantation with delayed total hepatectomy (RAPID) was introduced in 2015 [75]. For this approach, a two stage hepatectomy with transplantation of a segment 2/3 graft (potentially also in a living donor (LD) situation) in the first step aims to provide time for regeneration of the transplant, and completion hepatectomy in a second step is then performed. This novel concept is currently being investigated at different sites in ongoing studies, including in Oslo (Norway), Padova (Italy), Jena, Tübingen, and Munich (Germany) as summarized in Table 1. Of note, studies involving LT of CRLM, dynamic biological factors such as low levels of CEA, adequate response, and stability towards therapy as well as exclusion of genetic mutations with worse prognosis, such as BRAF-V600^MT^ or MSI/MMRd, are prerequisite conditions for study inclusion of these selected patients. Importantly, some of these protocols incorporate mTOR inhibition for the post-transplantation course with Sirolimus (e.g., SECA-I and SECA-II [70]) or everolimus in the RAPID-MUC trial, which might change postoperative oncologic courses in these patients as discussed below.

## 3. Immunosuppression and Cancer

### 3.1. Immunosuppression and Tumor Progression

The conventional wisdom is that high immune suppression should lead to more tumor recurrences. Chronic immunosuppression was shown to promote de novo tumors [76]. The effect of immunosuppression on established tumors that have already escaped immune surveillance at the time of their clinical appearance is less clear. There is some preclinical evidence that this might be the case, but in the clinical scenario, patients’ oncological outcomes are similar to those not immunosuppressed [77].

### 3.2. Antitumoral Immunosuppression with mTOR Inhibitors

It seems that immunosuppressants may exert different effects on tumor progression. In many experimental studies, an antitumor effect of mTOR inhibitors was shown [78]. The proposed mechanisms for antitumoral effects of mTOR inhibitors are manifold. In some tumors with activation of the mTOR pathway, mTOR inhibitors may exert direct antiproliferative effects. However, indirect effects of mTOR inhibitors on tumors probably appear to be more important. In this context, antiangiogenic, antivascular [79], and antilymphogenic effects [80] have been described for this class of immunosuppressants. Moreover, the immunologic system seems to be influenced by mTOR inhibitors differently than by calcineurin inhibitors.

Meanwhile, there are also a number of clinical studies demonstrating adjuvant effects of mTOR inhibitors on tumor recurrence. This is particularly true for tumors with poor tumor biology. A multivariate analysis of the SILVER trail showed that risk of recurrence in HCC with an AFP ≥10 ng/mL was reduced by half under the mTOR inhibitor sirolimus [81]. For optimal tumor control it seems to be important to avoid immunosuppressive escalations due to rejection episodes and to maintain instead effective but low immunosuppressive levels, preferable with an mTOR inhibitor-based regimen. In recent years, dual maintenance immunosuppression with low dose tacrolimus and everolimus has become widely accepted in tumor patients (Table 1).

### 3.3. Immunosuppression and Checkpoint Inhibitors

Immunotherapy, specifically immune checkpoint inhibitors (ICIs), including anti-programmed cell death 1 (anti-PD1), has recently received clinical approval for the treatment of adult HCC with durable responses in at least 25% of patients with advanced disease [82]. Due to the considerable risk of inducing an acute rejection reaction, ICIs can only be used with extreme caution. Rejection rates after organ transplantation and ICI therapy were reported to be as high as 36–60% [83]. However, recent case reports suggested that in selected cases and under stringent surveillance, use of ICIs such as the PD1 inhibitor Nivolumab might be feasible [84].

ICIs are likely to be more important in the neoadjuvant setting. Here, sustained response to therapy may serve as a positive selection criterion in a dynamic selection process.

A summary of ongoing studies regarding checkpoint inhibition in recurrent HCC after LT can be found in Table 2.

## 4. Prioritization of Patients with Oncologic Indications

Theoretically, liver transplantation is a viable therapeutic option for a variety of non-resectable primary liver tumors and liver metastases without extrahepatic disease. With increasingly effective systemic therapy, patients with non-resectable CRC-LM, in particular, could burden the donor pool and outnumber benign indications. This in turn raises the question how to prioritize a potentially large number of patients with oncological indications. The currently used MELD allocation system focuses on patients with end-stage liver disease, while patients with tumors are prioritized within this system via the assignment of standard exception points. In this respect, the balance between benign and malignant indication remains a challenge. A possible solution to the problem was proposed by Line et al. with the RAPID procedure for patients with CRLM, performing a two-stage LT with a left lateral liver segment [75]. This technique unwraps the possibility to maximize available post-mortal organs by using one organ for more than one recipient or bypass shortage and increase the donor pool with living donor transplantations.

A further problem may arise from the fact that in some tumor entities tumor recurrence rates after transplantation is fairly high, questioning the curative intent of these transplantations. This may be particularly true for patients with CRC and NET liver metastases. In liver transplantation for NET-LM, spontaneous disease course under alternative treatment modalities is excellent, jeopardizing the transplant benefit for this indication [69].

Risk stratification and assessment especially for post-LT outcome in HCC has been of long-standing interest for the abovementioned reasons. Biological criteria might further help to fine tune current approaches; however, more research in this regard will be necessary. Machine learning was shown to be feasible and highly accurate in calculating potential recurrence risk for HCC [85]. Further tools of artificial intelligence might be very useful to also assess usefulness, clinical outcome, and recurrence across malignant and benign indications for LT. Particularly in countries with high organ scarcity, this could help to find organ recipients with the highest level of benefit.

## 5. Future Perspective and Conclusions

In recent years, the determination of tumor biology based on clinical courses, response to therapy, and tumor markers has taken on an increasingly important role in the selection of transplant patients with malignancies, with morphometric criteria increasingly being abandoned. In the future, morphometric criteria will only represent the guard rails of technical resectability, while dynamic criteria based on tumor biology will be used for the individual selection of transplant candidates (Figure 2).

Antitumor immunosuppressive regimens with mTOR inhibition may reduce recurrence rates, but more than in the past, this goal must also be pursued with adjuvant chemo- and immunotherapies. As the costs for tumor genome sequencing substantially shrink, prices will be more amenable and genomic deciphering of primary tumor and/or liver metastasis of treatable mutations for subsequent individualized molecular therapies will play a huge role in pre- and posttransplant therapies. Moreover, liquid biopsies of circulating tumor DNA might be useful for post-transplantation surveillance and during immunotherapy or for pre-LT assessment of beneficial tumor biology [30].

The real limitation to expanding LT indications is organ shortage. Living donation and partial liver transplantation are not sufficient to meet the demand. It is possible that the availability of xenogeneic or artificial donor organs will offer new alternatives in the future.

With these issues largely resolved, transplantation might expand the surgical toolbox in the treatment of advanced and biologically appropriate hepatobiliary tumors.

## Figures and Tables

**Figure 1 cancers-14-02662-f001:**
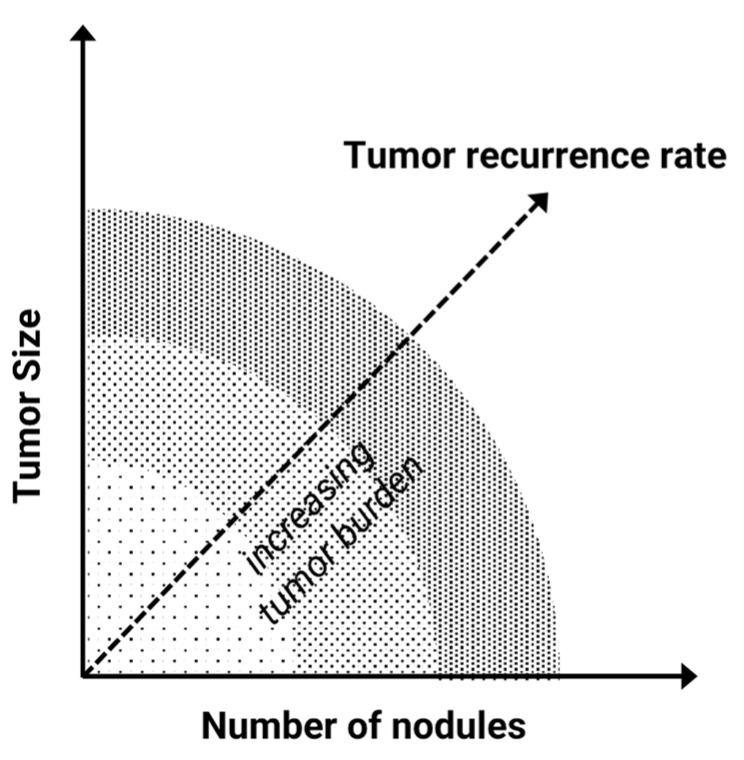
Traditional concept of selection criteria for transplant candidates by tumor burden as defined by number of nodules and tumor size (e.g., *Milan criteria*). Theoretically, according to this model, the tumor recurrence risk should incrementally increase according to the area under the curve representative for total hepatic tumor burden.

**Figure 2 cancers-14-02662-f002:**
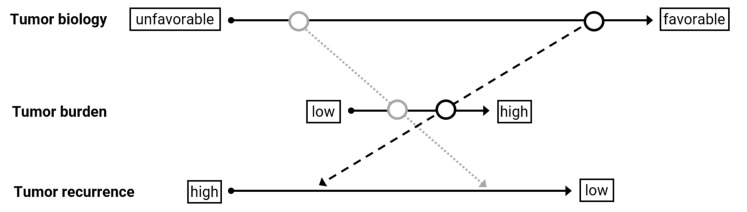
Individual tumor biology, within the limits of technical resectability, is more decisive for the tumor recurrence rate than mere morphometric criteria. Tumors with a low tumor burden but unfavorable tumor biology (arrow in light grey) may have worse outcome as compared to tumors with high tumor burden but favorable tumor biology (arrow in black).

**Table 1 cancers-14-02662-t001:** Ongoing (surgical) studies for LT in different pathologies.

Underlying Pathology	Clinical Trial ID	Acronym/Name	Design	Recruitment	Location	Phase
CCC	NCT02232932	TRANSPHIL	Prospective, randomized, multicenter study comparing neoadjuvant chemo-radiotherapy followed by LT and conventional operative resection for resectable phCCA	2024	France	N/A
	NCT04993131	TESLA-II	LT for non-resectable phCCA (prospective, exploratory)	2035	Norway	N/A
	NCT04378023		LT combined with neoadjuvant chemo-radiotherapy in the treatment of unresectable phCCA. A prospective multicenter study	2025	Spain	N/A
	NCT04556214	TESLA	LT for non-resectable iCCA: a prospective exploratory trial	2035	Norway	N/A
	NCT02878473		Single-arm, prospective, international multicenter study to evaluate the effectiveness of LT for very early iCCA (<2 cm) in cirrhotic patients (CA-19.9 <100 ng/mL)	2029	Canada	2
CRLM	NCT02864485		LD-LT for unresectable CRLM	2023	Toronto, Canada	
	NCT01479608	SECA-II	Open label, randomized controlled trial to assess the OS between patients undergoing LT or liver resection; deceased donor LT with liver resection in selected patients with six or more liver-only metastases from colorectal cancer deemed technically resectable	2027	Norway	3
	NCT02597348	TRANSMET	comparing LT after standard chemotherapy and standard chemotherapy alone for unresectable CRLM; the main outcome is 3- and 5-year DFS/PFS	2027	France	3
	NCT02215889	Resection and Partial Liver Segment 2/3 Transplantation with Delayed Total Hepatectomy (RAPID) Trial	The RAPID concept is to perform a left lateral segmentectomy and orthotopic transplantation of a left lateral segment graft. The total hepatectomy is delayed until the transplanted graft has reached sufficient volume. Liver resection and partial section 2/3 Tx w/two-stage hepatectomy	2028	Norway	1–2
	NCT04865471	RAPID-PADOVA	Resection and partial liver segmental transplantation with delayed total hepatectomy as treatment for selected patients with unresectable CRLM	2025	Italy	N/A
	NCT03488953	LIVERT(W) OHEAL	LDLT w/two-stage hepatectomy	2023	Germany	1/2
	DRKS00017730	RAPID-MUC	Partial segment 2/3 LT with two-stage complete hepatectomy as therapy for selected patients with CRLM		Munich, Germany	
	NCT03494946	SECA-III	LT vs. chemo or ablation	2027	Norway	3
	NCT02864485	Toronto Protocol	Chemo + LDLT vs. chemo	2023	Canada	
	NCT03803436	COLT	Chemo + LT vs. chemo	2024	Italy	
	NCT04161092	SOULMATE	Chemo + LT w/ECD vs. chemo	2029	Sweden	
	NCT04616495	TRASMETIR		2028	Spain	
Others	NCT04825470	TRANSGIST	LT for unresectable GIST LM	NYR 2021–2025	Spain	

**Table 2 cancers-14-02662-t002:** Bridging and immunotherapies in LT—ongoing studies.

Underlying Pathology	Clinical Trial ID	Acronym/Name	Design	Recruitment	Country	Phase
HCC	NCT04425226		Safety/efficacy of PD-1 inhibitor in combination with Lenvatinib as neoadjuvant therapy in patients with HCC			
	NCT01551212	HEPHAISTOS	Randomized to everolimus + TAC or TAC + MFF			
	NCT03966209		Evaluation of PD-1 inhibition in patients with recurrent HCC after LT	2019–2022	China	
	NCT04564313		Safety and Efficacy of Camrelizumab (anti-PD-1 antibody) in recurrent HCC after LT	2020–2023	China

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
