# Peer review of "Liver Transplant Oncology: Towards Dynamic Tumor-Biology-Oriented Patient Selection"

_cancers, 2022, doi:10.3390/cancers14112662_

Round 1

Reviewer 1 Report

This narrative review from Ilmer and Guba entitled “Liver Transplant Oncology - Towards a Dynamic Tumor Biology-oriented Patient Selection” provides an interesting summary of the current evidence concerning the evolution of selection criteria for liver transplant candidates presenting primary and secondary liver malignancies.

The subject is really wide and therefore making a concise review is particularly challenging, despite that, the Authors provided a good summary of the most recent literature.

The paper is well written, concise, and properly organized in different sections for each indication

There are some integrations that should be made in order to improve the paper quality:

HCC
1) While referring to tumor burden assessment, the Authors should include the recent evidence supporting a thoughtful application of LI-RADS protocol in preoperative assessment of LT candidate: 10.1016/j.aohep.2020.06.007; 10.1111/tri.13983
2) The Authors should speculate about the introduction of artificial intelligence into risk stratification: 10.1002/lt.26332

CCA
1) The Authors should include the recent publication form the Houston group providing the results for advanced CCA: 10.1111/ajt.16906
2) Add some comments concerning the comparisons of surgical resection vs. transplantation for advanced CCA, that has been recently challenged by Nagino: 10.1097/SLA.0000000000002624

Last, the Tables should be included in the manuscript

Congratulations to the Author for their interesting review

Best regards

Author Response

This narrative review from Ilmer and Guba entitled “Liver Transplant Oncology - Towards a Dynamic Tumor Biology-oriented Patient Selection” provides an interesting summary of the current evidence concerning the evolution of selection criteria for liver transplant candidates presenting primary and secondary liver malignancies.

The subject is really wide and therefore making a concise review is particularly challenging, despite that, the Authors provided a good summary of the most recent literature.

The paper is well written, concise, and properly organized in different sections for each indication

There are some integrations that should be made in order to improve the paper quality:

HCC
1) While referring to tumor burden assessment, the Authors should include the recent evidence supporting a thoughtful application of LI-RADS protocol in preoperative assessment of LT candidate: 10.1016/j.aohep.2020.06.007; 10.1111/tri.13983

Thank you for your feedback and the suggestion of the mentioned articles. At the end of the HCC chapter (2.1), we included the following paragraph:

Last, preoperative assessment of liver tumors and outcome after LT is still chal-lenging and sometimes, even the distinction between benign and malignant lesions remains to be a demanding task. In this regard, the use of the Metroticket 2.0 calculator might help to assess preoperative HCC tumor burden and biology; however, a better assessment applying the Liver Imaging Reporting and Data System (LI-RADS) protocol has been recently suggested [25]. In particular, LI-RADS4 and 5 highly correlated with HCC pathology in examinations of explanted livers [26]. In this setting, it would be de-sirable to establish extended criteria of LI-RADS that could help to categorize not only the likelihood of finding HCC, but also to evaluate biological features of HCCs, which could further help to estimate the use of LT as well as long-term prognosis. In this context, application and incorporation of artificial intelligence with a focus on radiomics in the process of finding the right candidate and maybe the right organ, could be of prime value in transplant oncology.

2) The Authors should speculate about the introduction of artificial intelligence into risk stratification: 10.1002/lt.26332

Thank you for this very interesting suggestion. We included a small paragraph speculating on the usefulness of AI, especially in the setting of limited organ supply, to help find the best organ recipient across benign and malignant indications for LT at the end of the HCC chapter as well as chapter 4.

CCA
1) The Authors should include the recent publication form the Houston group providing the results for advanced CCA: 10.1111/ajt.16906

Thank you for this suggestion. In chapter 2.2 paragraph 3, we included the following:

A recent update from the McMillan and colleagues showed that even advanced iCCAs with median tumor size of 10.4 cm and median number of nodules of 2 might benefit from LT. Here, favorable genetic alterations (e.g., in FGFR and DNA damage repair pathways) that associate iCCAs with lower aggressive behavior and superior response to neoadjuvant therapy might add increased value to the selection process. However, recurrence occurred nevertheless in almost 50% of transplanted patients.

2) Add some comments concerning the comparisons of surgical resection vs. transplantation for advanced CCA, that has been recently challenged by Nagino: 10.1097/SLA.0000000000002624

In paragraph 4 of chapter 2.2 we included the following:

However, these criteria remain a matter of debate. Recently, the (un-)resectability of phCCAs defined by the above-mentioned criteria has been challenged, while outcome (5-year OS) of resected patients in a different cohort was 67.1% questioning the advantage of LT over resection.

Last, the Tables should be included in the manuscript

Thank you for this comment. Tables have been included in the revised version of this manuscript.

Congratulations to the Author for their interesting review

Best regards

Reviewer 2 Report

Major Comments:

  1.  When comparing transplant outcomes for malignant indications in the setting of limited organ availability, it is critical to compare malignant transplant outcomes to non-malignant transplant outcomes.  The comparison of malignant transplant outcomes to non-transplant malignant outcomes is not appropriate.  This is most evident in section 2.6, penultimate paragraph where the authors state, "...patients with high tumor burden but low biological relevance might profit the most from LT compared to resection (5-year OS 69.1% vs. 14.6%, 1-yearDFS 54.2% vs. 11.5%).  These transplant outcomes, while superior to resection, are unacceptably low and may result in censoring of the transplant program.
  2.  The authors should be clear about these comparisons.  For example, in Section 2.1, 4th paragraph, final sentence: "HCCs fulfilling these criteria achieved excellent 5 year overall survival of 71.2%"  Did the authors mean with our without transplantation?  The discussion of the Lancet Oncology 2009 paper presents the same issue: "...presence of MiV was associated with a reduced 5-year survival of 47.4% compared to 71.2% without MiV."  Are these results in the setting of transplantation?
  3. In the final sentence of section 2.1, is there are p value?
  4. In the final paragraph of section 2.5: The comment that NETLM patients would require "excellent donor organs" in order to overcome the added mortality associated with "marginal donor organs" is speculative at best.  If true, the concept could be applied to any indication for transplantation.  The phrase "marginal donor organs" is no longer used in favor of "extended risk organs".
  5. In section 2.6, 3rd paragraph: In the statement, "Candidates for LT needed to respond to chemotherapy with at least 10%...."  10% of what?
  6. In the second to last sentence of section 4.2: Is there a reference for the claim that low immunosuppression including an mTOR inhibitor is is preferable for CRC-LM patients in particular?
  7. The final statement of the manuscript, "With these issues [living donation, xenotransplantation, artifical donor organs] largely resolved..." is grandiose and dismissive of the current knowledge and deliberate progress of field.

Minor Comments:

  1.  There are syntax errors throughout that can be addressed by the editorial staff.
  2. In section 2.2,  3rd paragraph and section 2.6 penultimate paragraph, would choose an alternative word to "profit" as this implies financial gain.
  3. In section 2.2, 4th paragraph, I believe the the authors mean "CA19-9 <1000 kU/L."
  4. Section 2.3, first paragraph: The acronym "CCC" has not been used previously and may be "CCA"?  NETLM should be spelled out the first time it appears in the manuscript.
  5. Tables 1 and 2 were not made available to this reviewer.

Author Response

Major Comments:

  1.  When comparing transplant outcomes for malignant indications in the setting of limited organ availability, it is critical to compare malignant transplant outcomes to non-malignant transplant outcomes.  The comparison of malignant transplant outcomes to non-transplant malignant outcomes is not appropriate.  This is most evident in section 2.6, penultimate paragraph where the authors state, "...patients with high tumor burden but low biological relevance might profit the most from LT compared to resection (5-year OS 69.1% vs. 14.6%, 1-yearDFS 54.2% vs. 11.5%).  These transplant outcomes, while superior to resection, are unacceptably low and may result in censoring of the transplant program.

First, we thank the reviewer for his careful analysis of our manuscript and the detailed comments. We also agree with the reviewer that on the one hand outcomes for oncological indications should be compared with benign indications, especially in the light of fair allocation. However, on the other hand, in oncological research it is standard to compare treated to untreated patients which would be the case in the papers that we cited. With this in mind, we would find it scientifically challenging to compare outcomes of different studies in order to reach a comparison of malignant vs. non-malignant indications.

  1.  The authors should be clear about these comparisons.  For example, in Section 2.1, 4th paragraph, final sentence: "HCCs fulfilling these criteria achieved excellent 5 year overall survival of 71.2%"  Did the authors mean with our without transplantation?  The discussion of the Lancet Oncology 2009 paper presents the same issue: "...presence of MiV was associated with a reduced 5-year survival of 47.4% compared to 71.2% without MiV."  Are these results in the setting of transplantation?

Thank you for pointing this out. In the referenced paper, all mentioned survival data refer to post-LT outcomes. We included corrections in the current version of the manuscript to clarify any misunderstandings.

  1. In the final sentence of section 2.1, is there are p value?

The p value was p=0.552 comparing inside MC vs. outside MC / inside MUC.

  1. In the final paragraph of section 2.5: The comment that NETLM patients would require "excellent donor organs" in order to overcome the added mortality associated with "marginal donor organs" is speculative at best.  If true, the concept could be applied to any indication for transplantation.

We agree with your assessment. To avoid any misunderstandings, we took this section out of the current version of the manuscript. 

The phrase "marginal donor organs" is no longer used in favor of "extended risk organs".

We apologize for the wording and changed it to extended risk organs.

  1. In section 2.6, 3rd paragraph: In the statement, "Candidates for LT needed to respond to chemotherapy with at least 10%...."  10% of what?

Thank you for this remark. In the original paper it was described as follows:

“The patients had significant higher tumor load at the time of diagnosis than before LT. The difference between maximal CEA, size of largest liver metastases, number of liver lesions and FCRS, and values at time of LT were significantly different with P values of 0.001, 0.003, 0.001, and 0.014 for a change in CEA levels, number of lesions, and size of liver metastases and FCRS, respectively. These differences indicated that the patients had response to chemotherapy and one of the inclusion criteria for the study was at least 10% response on chemotherapy at the time of LT according to RECIST criteria.”

Hence, we included the following wording in this paragraph to clarify what is meant:

Candidates for LT needed to respond to chemotherapy with at least 10% according to RECIST criteria from the point of diagnosis to the time for LT and time from diagnosis to LT mandatorily exceeded 1 year.

  1. In the second to last sentence of section 4.2: Is there a reference for the claim that low immunosuppression including an mTOR inhibitor is is preferable for CRC-LM patients in particular?

Thank you for pointing this out. We don’t have evidence to support this claim and took the sentence out.

  1. The final statement of the manuscript, "With these issues [living donation, xenotransplantation, artifical donor organs] largely resolved..." is grandiose and dismissive of the current knowledge and deliberate progress of field.

After re-reading this sentence, we find that we didn’t bring across the point that we wanted to make appropriately and rewrote it. It now says:

“With these issues largely resolved, transplantation might expand the surgical toolbox in the treatment of advanced and biologically appropriate hepatobiliary tumors.”

Minor Comments:

  1.  There are syntax errors throughout that can be addressed by the editorial staff.
  2. In section 2.2,  3rd paragraph and section 2.6 penultimate paragraph, would choose an alternative word to "profit" as this implies financial gain.

We exchanged “profit” to “benefit”.

  1. In section 2.2, 4th paragraph, I believe the the authors mean "CA19-9 <1000 kU/L."

Thank you for pointing out this mistake. Diagnosis should be made by pathologic confirmation or radiologically malignant-appearing stricture with a CA 19-9 level > 100 U/mL and/or FISH polysomy (http://dx.doi.org/10.1053/j.gastro.2013.10.013).

This was corrected in the text.

  1. Section 2.3, first paragraph: The acronym "CCC" has not been used previously and may be "CCA"?  NETLM should be spelled out the first time it appears in the manuscript.

Thank you for pointing this out. Both abbreviations have been changed/modified according to your suggestions.

  1. Tables 1 and 2 were not made available to this reviewer.

Tables 1 and 2 are now available in the manuscript.

Reviewer 3 Report

In this review, the authors discuss present-day oncologic indications for liver transplantation and reviewed the currently applied and evolving selection criteria for oncologic liver transplant patients, implications for (neo-)adjuvant therapy, and discuss the allocation equity issues that an expansion of oncologic indications in liver transplantation may entail.

The review is of interest and of clinical relevance.

However, the review could be further improved by discussing the problem of HCC recurrence after LT. In particular, data on the outcome of sorafenib and regorafenib treatments are now available and should be discussed as recently reported (Managements of recurrent hepatocellular carcinoma after liver transplantation: A systematic review. World J Gastroenterol. 2015 Oct 21;21(39):11185-98; Experience with regorafenib in the treatment of hepatocellular carcinoma. Therap Adv Gastroenterol. 2021 May 28;14:17562848211016959).

Author Response

In this review, the authors discuss present-day oncologic indications for liver transplantation and reviewed the currently applied and evolving selection criteria for oncologic liver transplant patients, implications for (neo-)adjuvant therapy, and discuss the allocation equity issues that an expansion of oncologic indications in liver transplantation may entail.

The review is of interest and of clinical relevance.

We thank the reviewer for the positive feedback of our manuscript as well as the suggestions for further improvement. Below, you can find our discussions regarding the suggested literature.

However, the review could be further improved by discussing the problem of HCC recurrence after LT. In particular, data on the outcome of sorafenib and regorafenib treatments are now available and should be discussed as recently reported (Managements of recurrent hepatocellular carcinoma after liver transplantation: A systematic review. World J Gastroenterol. 2015 Oct 21;21(39):11185-98; Experience with regorafenib in the treatment of hepatocellular carcinoma. Therap Adv Gastroenterol. 2021 May 28;14:17562848211016959).

The following paragraph incorporating your suggestions was inserted at the end of chapter 2.2.:

Recurrence of HCC after LT occurs in about 16% of patients; here, biological factors also determine the prognosis, e.g., early recurrence (potentially, due to circulating tumor cells (CTC) or undetected metastasis) vs. late recurrence (either by better containment of CTCs and occult metastasis or by de novo tumors in the newly grafted liver) as well as localized and isolated vs. multifocal (metastatic) disease. Surgical resection in these cases appears to be advantageous; theoretically, re-transplantation might be an option for biologically selected patients. If patients can’t be resected, loco-regional therapy is the best option and systemically, a combination of sorafenib with mTOR inhibitors improves patient survival (World J Gastroenterol. 2015 Oct 21;21(39):11185-98). Sorafenib-tolerant patients could further be treated by regorafenib, a multikinase inhibitor similar to sorafenib with a wider range of kinase inhibition modulating the tumor microenvironment and potentially promoting anti-tumor immunity. This might then extend overall survival in patients after HCC recurrence after LT without the option to resect (Therap Adv Gastroenterol. 2021 May 28;14:17562848211016959).

Round 2

Reviewer 1 Report

The Authors properly revise their original manuscript

The paper is now suitable for publication

Reviewer 2 Report

The manuscript has been significantly improved.